# OpenReview forum: "Torque-Aware Momentum"
_TMLR — Rejected by TMLR_

### Review · Reviewer_gsxE · 2025-08-21

**Summary Of Contributions:**

This paper introduces Torque-Aware Momentum (TAM), an enhancement to classic momentum-based optimizers. TAM adjusts momentum according to the cosine similarity between the current gradient and the previous momentum, reducing abrupt directional shifts and promoting stable exploration of the loss landscape. The authors demonstrate empirical improvements for TAM (and its adaptive variant AdaTAM) for various tasks—including CIFAR image classification, ImageNet fine-tuning, large language model tuning, and distribution-shift robustness.

**Additional Comments:**

N/A

**Audience:**

Yes

**Audience Explanation:**

Yes. The findings of this paper would be of particular interest to researchers working in deep learning, as the proposed Torque-Aware Momentum (TAM) offers a simple yet effective way to stabilize and improve training across diverse tasks.

**Claims And Evidence:**

Yes

**Claims Explanation:**

Yes. This paper proposes a new momentum-based optimizer and provides extensive experiments to demonstrate its effectiveness.

**Requested Changes:**

## Strengths
- **Novel idea**: The paper introduces Torque-Aware Momentum (TAM), which adjusts updates by damping gradients misaligned with the momentum direction. Using cosine similarity between gradient and momentum to guide updates is conceptually elegant and practically simple.
- **Broad empirical validation**: The method is evaluated across diverse tasks, including CIFAR classification, ImageNet fine-tuning, LLM fine-tuning, and distribution-shift robustness. Results consistently show TAM performs well compared to standard baselines.
- **Practical compatibility**: TAM integrates seamlessly with both SGD with momentum and Adam, requiring only a minor modification to existing optimizers.

---

## Weaknesses / Questions
1. **Choice of $\gamma$ (Equation 3)**
   In equation 3, the paper states that $\gamma$ is intended to be a decaying stepsize to account for stochasticity, yet also mentions that $\gamma=0.9$ is used as a fixed default. This discrepancy needs clarification. While experiments suggest varying $\gamma$ has little effect, more high-level intuition about *why* the algorithm is so robust to this choice would strengthen the explanation.

2. **Interpretation of Figure 3** ($s^t \to s^* = 0$)
   The figure shows the momentum of cosine similarity converging to 0. Since $s$ represents the cosine between momentum and gradient, $s=0$ implies near-orthogonality. (Please correct me if I am wrong.) This seems counterintuitive because momentum-based methods typically align momentum with the gradient direction as training stabilizes. Could the authors provide intuition or theoretical explanation for why $s^*$ tends to 0 in TAM?

3. **Role of the damping term across training**
   It seems the damping effect is strongest at the beginning of training, while in the later stages (where $s^* \approx 0$)) TAM updates resemble those of SGDM with a rescaled stepsize. If I understand correctly, then TAM’s main benefit is early-phase stabilization. More discussion (or theoretical guarantees) about how TAM shapes convergence behavior in the first phase of training would be valuable.

---

> ### Author Response · Authors · 2025-09-17
>
> Thank you for your thoughtful review and positive feedback. We appreciate your recognition of the strengths of our work, particularly the novelty of TAM, its broad empirical validation across diverse tasks, and its practical compatibility with existing optimizers. We address the concerns below:
>
> ------
>
> **Choice of $\gamma$ (Equation 3)**
>
> Thank you for pointing this out. To clarify, $\gamma$ in Eq. 3 is not a decaying stepsize, but a fixed smoothing factor that computes an exponential moving average of the cosine similarity.
>
> Our ablation in Section 4.6.1 (Table 3) shows that while varying \gamma has little effect on average performance, we also note that higher values (e.g., $0.9$ or $0.99$) noticeably reduce the variance in overall performance, which can be particularly important in non-stationary settings such as online or continual learning. Therefore, we used $\gamma=0.9$ by default in our main experiments. We have clarified this point in the revised manuscript.
>
> ------
>
> **Interpretation of Figure 3**
>
> You are correct that $\hat{s}_t \to 0$ implies near-orthogonality between momentum and the current gradient. While this may initially seem counterintuitive, it is a consequence of how gradients behave as training progresses into sharper regions of the loss surface. In these regions, gradients fluctuate more strongly around sharp minima, leading to oscillatory behavior. Since momentum averages past gradients, the new gradient tends to decorrelate from the accumulated direction and becomes nearly orthogonal in expectation.
>
> This observation is consistent with the “abrupt sharpening” phenomenon described by Fu et al. (2023), where gradients start to oscillate as the loss surface sharpens. TAM does not attempt to enforce alignment in this regime; rather, it stabilizes updates by damping the influence of misaligned gradients while still leveraging accumulated momentum. As shown in Figure 8, this delays this abrupt sharpening, allowing for longer, more consistent exploration and ultimately better generalization.
>
> We have added this interpretation to the manuscript to clarify that near-orthogonality is expected in this phase of training and does not imply loss of useful momentum information.
>
> ------
>
> **Role of the damping term across training**
>
> Yes. TAM’s damping is indeed most influential during the early phase of training, when gradient–momentum misalignment is frequent and can cause instability. This behavior is by design — TAM was specifically motivated to stabilize this phase and enable more consistent exploration of the loss landscape before the optimizer commits to a sharp minimum.
>
> Our analysis in Section 4.5 (Figure 8) shows that TAM delays the gradient-norm spike associated with the “abrupt sharpening” event (Fu et al., 2023) by 5–10 epochs, effectively extending the stable exploration phase. After this transition, TAM behaves similarly to SGDM with a rescaled step size (Eq. 6), preserving SGDM’s asymptotic convergence behavior. We have made this early-phase stabilization role more explicit in the revision and highlighted it as a primary contribution of TAM.
>
>
>
> ----------------------
>
>
>
>
> **We appreciate the review and its valuable feedback. We hope our responses address the concerns raised. Please feel free to reach out with any further questions, and we will respond promptly.**

---

### Review · Reviewer_wLkH · 2025-08-29

**Summary Of Contributions:**

This paper proposes a new optimization algorithm called _TAM_, which is based on a specific update of the momentum term (e.g., in the SGD+momentum algorithm). The design of TAM is based on the following observation: when performing SGD+momentum, it happens that the gradient is severly misaligned with the current momentum, which may slow the optimization process (if the momentum is already in the right direction).

Stated formally, the $t$-th step of the algorithm is, given a gradient $g_t$ and a momentum $m_{t-1}$:
* compute the cosine similarity between $g_t$ and $m_{t-1}$: $S_t = m_{t-1} \cdot g_t / (\|m_{t-1}\| \, \|g_t\|)$;
* compute the exponential moving average of $S_t$: $\hat{s}\_t = \gamma \hat{s}_{t-1} + (1 - \gamma) S_t$, where $\gamma = 0.9$;
* scale $\hat{s}_t$ to make it belong to $[0, 1]$: $d_t = (1+\hat{s}_t)/2$;
* update the momentum: $m_t = \beta m_{t-1} + (d_t + \epsilon_t) g_t$, where $\beta = 0.9$ and $\epsilon = 10^{-8}$;
* update the parameters: $\theta_t = \theta_{t-1} - \eta m_t$.

**Audience:**

Yes

**Audience Explanation:**

Proposing a new variant of momentum is relevant for the community.

**Empirical findings.**
Figure 3 is surprising: it seems to show that the $g_t$ are mostly negatively correlated to the $m_{t-1}$. It would be less surprising if they were orthogonal. This kind of finding can be very interesting and should be investigated further.

**Claims And Evidence:**

No

**Claims Explanation:**

For the reason below, the significance of the work is difficult to evaluate.

**Methodological concerns (experiments).**
According to the article, a grid search is performed on the learning rate, and the validation accuracies are reported. My concern is about the number of splits of the datasets: are there 2 (training+valid) or 3 (training+valid+test) subsets? It has implications on the results of the experiments:
1. Which criterion is used to pick the best learning rate? The validation accuracy? If it is, and if the validation accuracy is also used in the final experiments, the setup is not fair: it is possible that the "best learning rate" chosen is adapted only to a specific subset of the dataset.
2. Is early stopping used? It would be advisable to look for the best validation accuracy _during_ training, which may be better that _at the end_ of training. Why checking that? Because it is posible that the SGD is prone to overfitting at the end of training, while TAM is not, but the SGD achieves better accuracy at the middle of the training. In this situation, we would mitigate the relevance of TAM.

Additionally, would it be possible to provide some training curves? (with the training+valid accuracies)

**Relevance of the hyperparameter $\gamma$.**
Taking $\gamma > 0$ is computing the exponential moving average of $S_t$. Why? what is the purpose? If the goal is to mitigate the influence of the current $g_t$, then one should only take into account the current $g_t$ in the computation, that is, use directly $S_t$, and ignore $\hat{s}\_t$. If, for stability reasons (or other), the $S_t$ need to be averaged, the initial motivation (do not take much into account $g_t$ if it is misaligned with $m_{t-1}$) is weakened.

Moreover, the authors provide additional experiments in Section 4.6.1, which show that setting $\gamma$ to $0.99$ or to $0$ (i.e., $\hat{s}_t = S_t$) has basically no impact on the final performance when training ResNet18 on CIFAR10 and CIFAR100. Therefore, it is surprising that the computation of $\hat{s}_t$ (i.e., taking $\hat{s}_t = S_t$) has not been removed from the algorithm. If the authors encountered experiments where $\gamma = 0.9$ led to much better results than $\gamma = 0$, it should be mentioned somewhere (and it would contradict the claim that the value of the hyperparameter $\gamma$ is unimportant).

**Why $\hat{s}_t < 0$?**
Although Figure 3 is interesting, it is under-studied in this article. For instance: with SGD+momentum, do we observe in general that $g_t$ and $m_{t-1}$ are negatively correlated?

**Requested Changes:**

Clarify the train/valid/test split of the dataset: which accuracy is used during the grid search? what is the effect of early stopping on the results?

Clarify why $\gamma = 0.9$ is used, although Section 4.6.1 shows that $\gamma = 0$ would be good.

Add some explanations about Figure 3 (this figure can even be highlighted as an interesting empirical finding): the results are surprising and deserve some comments.

---

> ### Author Response · Authors · 2025-09-17
>
> Thank you for your detailed and thoughtful review. We appreciate your recognition of the relevance of our new variant of momentum. We address the concerns below:
>
> ------
> **Methodological concerns (experiments)**
>
> Thank you for pointing this out. For all cases except for ImageNet, we've used training+valid+test splits. For CIFAR10/100, the training set was further split into 90\%/10\% for training/validation, and the reported results are on the test set, which contains 10K samples. For ImageNet, we report results on the validation set. For MovieLens, the data is split into 80\%/10\%/10\% for training/validation/test. In all cases, the validation set is used to select the best learning rate. We use early stopping for selecting the checkpoint with the highest validation accuracy, ensuring a fair and unbiased evaluation. We've added these details in Appendix (A.1.1).
>
> In response to the reviewer’s suggestion, we have also included in the revision curves corresponding to training loss and test accuracy for SGDM vs. TAM on CIFAR10+ResNet18 (A.2.1). The results highlight the stabilizing role of TAM compared to SGDM during early-phase training and also show how TAM achieves higher accuracy across training steps.
>
> --------
>
> **Relevance of the hyperparameter $\gamma$**
>
> The role of $\gamma$ in Eq. 3 is to smooth the cosine similarity signal and reduce the impact of noisy mini-batch fluctuations. Our ablation in Section 4.6.1 (Table 3) shows that while varying $\gamma$ has little effect on average performance, we also note that higher values (e.g., $0.9$ or $0.99$) noticeably reduce the variance in overall performance, which can be particularly important in non-stationary settings such as online or continual learning. Therefore, we used $\gamma=0.9$ by default in our main experiments. We have clarified this point in the revised manuscript.
>
> -------
>
> **Why $\hat{s}_t < 0$?**
>
> We agree that Figure 3 reveals an interesting phenomenon. Our interpretation is that as training progresses, the momentum term continues to push parameters along previously dominant directions, while new gradients, influenced by increasing curvature, fluctuate more strongly and decorrelate from past directions. This decorrelation drives the cosine similarity toward zero, and in some cases negative, indicating that fresh gradients occasionally oppose the accumulated momentum.
>
> This behavior aligns with the abrupt sharpening effect described by Fu et al. (2023), where gradients begin to oscillate as the optimizer approaches sharp minima. In this view, the negative correlation reflects the optimizer resisting curvature-induced oscillations. TAM’s damping mechanism mitigates the influence of these misaligned gradients, delaying the sharpening event and allowing longer exploration of flatter regions. (Section 4.5, Fig. 8). We have added this explanation to the main text and explicitly highlighted Figure 3 as an empirical observation of interest that motivates TAM.
>
>
> ---------------
>
> **We appreciate the review and its valuable feedback. We hope our responses address the concerns raised. Please feel free to reach out with any further questions, and we will respond promptly.**

---

> > ### Comment · Reviewer_wLkH · 2025-09-18
> >
> > Thank you for your answer.
> >
> > ## Training curves
> >
> > Figure 10 : "The left subplot shows the evolution of train loss, highlighting the faster and more stable convergence of TAM compared to SGDM."
> >
> > As a matter of fact, the _training_ loss of TAM starts to increase significantly at the 90th epoch. SGDM also suffers from an increase, but it is much smaller. Therefore, one cannot say that the convergence of TAM is more stable. Actually, the training dynamics of TAM is extremely strange: having a training loss that increases consistently at some point tends to show that this training algorithm is not reliable.
> >
> > ## Relevance of $\gamma$
> >
> > The ablation study, in particular Table 3, is not very convincing. The authors admit that there is no significant difference of accuracy between the different choices of $\gamma$, but they argue that "higher values [of $\gamma$] noticeably reduce the variance in overall performance". This claim is not supported by Table 3: for both CIFAR-10 and CIFAR-100, the largest variances in the results are observed for intermediary values of $\gamma$, not for $\gamma = 0$. This is striking in the case of CIFAR-10 (variance of $0.1$ both for small and large values of $\gamma$, variance of $0.2$ otherwise). These inconsistencies show that the phenomenon is not fully understood by the authors, and that we are lacking consistent empirical evidence that the choice of $\gamma = 0.9$ is actually good in most situations.
> >
> > Moreover, taking $\gamma > 0$ may be the source of an unwanted momentum in the training process, which would cause the strange phenomenon of increasing training loss mentioned above.
> >
> > ## Required changes
> >
> > The consistency between the claims/explanations and the actual results should be checked thoroughly. I would recommend the authors to set $\gamma = 0$ and check again if their experimental results remain the same. If so, the proposed algorithm can be simplified, no ablation study would be needed, and the message would become clearer.

---

> ### Author Response · Authors · 2025-09-18
>
> Thank you for your response.
>
> **Training curves**
>
> We'd like to point the reviewer to Section 3.3, where we've mentioned that TAM’s advantage lies primarily in the early phase. In later stages, as $s^t \to 0$, TAM behaves like SGDM, which explains why their curves align. We'd also like to clarify that the apparent increase in TAM’s loss after ~90 epochs is partly visual due to the *log-scale* y-axis. In absolute terms, the magnitude of the fluctuations is comparable to that of SGDM, indicating that these late-phase behaviours are similar for both optimizers and do not reflect instability unique to TAM.
>
>
> **Relevance of $\gamma$**
>
> We respectfully disagree. While CIFAR-10 shows similar variance across values, CIFAR-100 clearly shows larger variance for $\gamma=\{0, 0.5, 0.8\}$ ($=\{0.3, 0.4, 0.3\}$ as compared $0.1$ for higher $\gamma$). We've clarified this point further in the paper. Regarding the concern that $\gamma>0$ might introduce unwanted momentum and cause the training loss to increase: we do not believe this is the case, as SGDM (which does not use $\gamma$) shows a similar late-phase increase in training loss.
>
> While rerunning all experiments with $\gamma=0$ would not be feasible, we are currently conducting additional experiments to further validate this effect, and will update the paper with these results once completed.
>
> --------
> We again appreciate the review and its valuable feedback. We hope our responses address the concerns raised. Please feel free to reach out with any further questions, and we will respond promptly.

---

### Review · Reviewer_WAhM · 2025-09-03

**Summary Of Contributions:**

## Summary
The paper introduces Torque-Aware Momentum (TAM), a simple modification to momentum-based optimizers. TAM scales the gradient by a damping factor determined by its alignment with past momentum, aiming to stabilize updates and mitigate oscillations. The method is easy to implement, theoretically connected to an effective learning rate interpretation, and evaluated across vision, language, and online learning tasks.

## Strengths
- The proposed method is conceptually simple yet well-motivated.
- The paper offers a heuristic theoretical interpretation and validates the approach across multiple tasks.

## Weaknesses
- The paper assumes oscillations are harmful but does not provide evidence; in some cases, oscillations may aid exploration.
- Performance gains on vision benchmarks are marginal and thus of limited practical significance; moreover, the language experiments would be stronger if evaluated on larger and more contemporary models, such as the GPT series or LLaMA.
- The theoretical analysis is heuristic and lacks formal convergence guarantees.
- Figure 7 shows sudden accuracy drops for TAM in some tasks, which calls for further explanation and raises questions about its claimed stability advantage over SGDM.

**Audience:**

Yes

**Audience Explanation:**

Momentum is a core technique in deep learning optimizers, and a novel and potentially effective variant like TAM is meaningful and of interest to the community.

**Claims And Evidence:**

No

**Claims Explanation:**

The claims are not fully convincing, as the evidence shows only marginal improvements and lacks rigorous theoretical and empirical support.

**Requested Changes:**

Please refer to the weaknesses.

---

> ### Author Response · Authors · 2025-09-17
>
> Thank you for your review. We appreciate your recognition of the strengths of our proposed method, particularly about being conceptually simple yet well-motivated, and that it offers a heuristic theoretical interpretation while validating the approach across multiple tasks. We appreciate the suggestions and have provided our responses below:
>
> ------
>
> **The paper assumes oscillations are harmful**
>
> Our objective is not to suppress all oscillations but to mitigate abrupt gradient–momentum misalignment (torque) that has been shown to slow convergence and impede effective exploration. Prior work (such as Fu et al., 2023; Ortiz-Jiménez et al., 2022) demonstrates that such oscillations are linked to the abrupt sharpening event, which increases update instability and can trap the optimizer in sharp minima.
>
> Our results directly illustrate the benefit of damping these events. In Section 4.5 (Figure 8) and Appendix A.2.1 (Figure 10), we show that TAM delays abrupt sharpening by 5–10 epochs, enabling a longer and more consistent exploration phase before settling. This behavior translates into improved final accuracy and smoother sharpness trajectories compared to SGDM.
>
> Thus, while mild oscillations may indeed promote exploration, TAM selectively mitigates the impact of mainly large, misaligned updates that destabilize training. After this early stabilization, TAM behaves similarly to SGDM (Section 4.5), allowing natural oscillations to re-emerge and contribute to exploration when beneficial.
>
> ------
>
> **Performance gains**
>
> While the absolute improvements on standard vision benchmarks such as CIFAR and ImageNet are indeed modest (0.3–0.7%) in Table 1, they are consistent across diverse architectures (MobileNet, ResNets, ViT) and achieved without additional hyperparameter tuning. Importantly, TAM’s benefits become much more pronounced in settings where stability is critical, such as distribution shift (Figure 7) and continual learning (Table 2), where TAM consistently outperforms SGDM by clear margins across all tasks.
>
> For language models, we emphasize that TAM’s adaptive variant, AdaTAMW,  improves generalization over AdamW across six BERT-family models (with number of parameters ranging from 110 to 355M) on a majority of the 56 MTEB datasets (Figures 4 and 5) and reduces validation perplexity, demonstrating scalability beyond vision tasks.
>
> In response to the reviewer’s suggestion, we also conducted a new experiment comparing  AdamW and AdaTAMW on GPT training. Using a grid search over three learning rates: {0.006, 0.0006, 0.00006}, we find that while both optimizers achieve similar validation loss by 100k iterations, confirming that their stabilization effect transfers to larger, more contemporary models.
>
> These new results have been added to the revised manuscript (A.2.3).
>
> | Iter    | AdamW | AdaTAMW |
> |---------|-------|---------|
> | 1       | 10.9  | 10.9    |
> | 1k      | 4.51  | 4.08    |
> | 5k      | 3.37  | 3.32    |
> | 25k     | 3.11  | 3.10    |
> | 100k    | 3.01  | 3.00    |
>
> ------
>
> **Theoretical analysis**
>
> We agree that our analysis is primarily heuristic and does not constitute a formal convergence proof. Our goal in Section 3.3 was to provide intuition by relating TAM’s update rule to SGDM’s through an effective learning rate equivalence (Eq. 6). This argument relies on the observation (Figure 3) that the cosine similarity stabilizes as training progresses, which implies that TAM’s effective learning rate remains bounded and tracks SGDM’s. Under these conditions, TAM inherits SGDM’s well-known convergence guarantees (Yan et al., 2018; Liu et al., 2020). We’ve highlighted this connection to SGDM’s theory in Section 3.3.
>
> ------
>
> **Figure 7**
>
> The temporary performance drops visible in Figure 7 are a direct consequence of the extreme distribution shifts in online learning, where all optimizers experience abrupt performance changes when a new task is introduced. These drops, therefore, reflect the task transition itself rather than the instability of TAM.
>
> Crucially, TAM exhibits significant recovery and higher sustained accuracy than SGDM after these transitions, particularly in the most challenging settings (δ = 80% and 100%), where it maintains superior performance over a long sequence of tasks. We have clarified this interpretation in the revised text to emphasize that the observed drops are expected and that TAM’s stability advantage lies in its quicker adaptation and better long-term performance.
>
> ------
>
> **We appreciate the review and its valuable feedback. We hope our responses address the concerns raised. Please feel free to reach out with any further questions, and we will respond promptly.**

---

> > ### Comment · Reviewer_WAhM · 2025-10-08
> > **Response to author**
> >
> > Many thanks for your detailed response. It has resolved many of my concerns and provided much-needed clarity.

---

### Decision · Action_Editor_3HFZ · 2025-10-15

**Recommendation:** Reject

**Additional Comments:**

I recommend a major revision of this paper.

Rather than focusing on more empirical evaluations (which are infeasible in many cases), I'd like to suggest improving the manuscript in terms of showing the cases when the original momentum fails, but TAM works. It is good to be a very simple toy example or synthetic loss surface, as many optimizer papers do.

More minor opinions might include (1) more theoretical results, such as convergence analysis, will improve the submission, but might not be mandatory, considering the TMLR criteria, and (2) including more recent optimizer papers (focusing on momentum) would be largely beneficial, but not necessary

**Audience:**

Yes

**Audience Explanation:**

Momentum-based optimizers are widely used optimizers in many machine learning applications. Pointing out the limitations of momentum optimizers might be interesting to many audiences.

**Claims And Evidence:**

No

**Claims Explanation:**

The main claim of this paper is that

1. Momentum could be vulnerable to the adverse effects of large, misaligned gradients
2. The proposed Torque-Aware Momentum (TAM) properly solves the problem
3. TAM actually shows better results than the standard momentum

In terms of evidence, this paper does not provide explicit evidence of (E1) the case when the standard momentum fails due to the misaligned gradients, (E2) the proposed TAM can actually solve the problem, and (E3) TAM satisfies the fundamental properties of optimizers (e.g., convergence guarantee, convergence complexity compared to the existing optimizers, ...)

I think that E1 and E2 are more critical because the main claim of this paper lies in the pitfall of the conventional momentum. In terms of TMLR criteria, some factors in E3 are optional; especially, "TAM is better than SGD" does not necessarily need to be supported. The convergence guarantee looks essential, and the current argument in Section 3.3 is not fully sufficient, but can be considered as an "Okay" support for TMLR.

As pointed out by the reviewers, the proposed method does not seem to reliably solve the proposed problem. For example, as pointed out by Reviewer wLkH, "the training curves do not decrease monotonically with the epochs (they even increase at some point), which shows that the dynamics of the method is not well controlled or understood".

The rebuttal clarifies that "TAM’s advantage lies primarily in the early phase", but I think that this cannot address Reviewer wLkH's concern. Even though TAM is beneficial in the early phase, it does not mean that TAM will work incorrectly in the later phase, which is observed in Figure 10 (and maybe in 40% flipped 4-MLP layers Figure 7? I'm not fully sure about this). It may support that TAM could lead to a suboptimal optimization process due to its fixed momentum operation. Namely, we need a new theoretical guarantee, or toy examples, when TAM actually converges better than the standard momentum. Now, such evidence is absent.

Additionally, I partially agree with Reviewer gsxE and wLkH, the concern about the hyperparameter. I don't think removing $\gamma$ is necessary, and the current version is somewhat acceptable (e.g., we don't test all $\beta_1, \beta_2$ of Adam variants). I don't think this is a major reject case for TMLR, but I'd like to mention that there is a fundamental potential limitation on the choice of the hyperparameter.

**Resubmission Of Major Revision:**

The authors may consider submitting a major revision at a later time.